

# *In situ* earthworm breeding in orchards significantly improves the growth, quality and yield of papaya (*Carica papaya* L.)

Huimin Xiang, Jia-en Zhang, Lei Guo and Benliang Zhao

College of Natural Resources and Environment, South China Agricultural University, Guangzhou, China
Guangdong Provincial Engineering Technology Research Center of Modern Eco-agriculture and Circular Agriculture, Guangzhou, China
Key Laboratory of Agro-environment in the Tropics, Ministry of Agriculture, South China Agricultural University, Guangzhou, China

## ABSTRACT

The aim of this study was to compare the effects of four fertilizer applications—control (C), chemical fertilizer (F), compost (O), and *in situ* earthworm breeding (E)—on the growth, quality and yield of papaya (*Carica papaya* L.). In this study, 5 g plant$^{-1}$ urea (CH$_4$N$_2$O, %$N$ = 46.3%) and 100 g plant$^{-1}$ microelement fertilizer was applied to each treatment. The fertilizer applications of these four treatments are different from each other. The results showed that the E treatment had the highest growth parameters over the whole growth period. At 127 days after transplantation, the order of plant heights from greatest to smallest was E > F > O > C, and the stem diameters were E > F > O > C, with significant differences between all treatments. Soluble-solid, sugar, vitamin C, and protein content significantly increased in the E treatment. In addition, the total acid and the electrical conductivity of the fruit significantly decreased in the E treatment. Fruit firmness clearly increased in the O treatment, and decreased in the F treatment. The fresh individual fruit weights, fruit numbers, and total yields were greatly improved in the F and E treatments, and the total yield of the E treatment was higher than that in the F treatment. In conclusion, the *in situ* earthworm breeding treatment performed better than conventional compost and chemical fertilizer treatments. Furthermore, *in situ* earthworm breeding may be a potential organic fertilizer application in orchards because it not only improves the fruit quality and yield but also reduces the amount of organic wastes from agriculture as a result of the activities of earthworms.

## INTRODUCTION

Papaya (*Carica papaya* L.) is one of the most important fruit crops which is widely cultivated in tropical and subtropical areas. It is rich in nutrition, sugar, vitamin C, protein, and amino acids, and it is the primary raw material that contains papain. Papaya is also widely planted in southern China, especially in Guangdong, Yunnan, and Hainan Provinces. However, papaya production is frequently low and unreliable. Although chemical fertilizer application is a common method for improving papaya yields, it is unfriendly to the environment. Chemical fertilizer can impair soil structure, and decrease soil fertility by reducing the carbon and nitrogen content (*Ngo et al., 2012*). Moreover, chemical fertilizer application

Corresponding author
Jia-en Zhang, jeanzh@scau.edu.cn

can also affect animal and human health (*Vu, Tran & Dang, 2007*) (for example, by killing some of the fish in rice paddies) and is bad for human health through the food chain due to the content of heavy metals in chemical fertilizer. For these reasons, the importance of organic fertilization has been increasing in recent years, and suitable organic amendments including composting, vermicomposting and *in situ* earthworm breeding have become promising biological ways to improve the growth, fruit quality and yield of papayas.

The addition of compost to soil has been described as an ideal alternative method for improving soil fertility and plant nutrition (*Cantanazaro, Williams & Sauve, 1998*; *Caravaca et al., 2002*), and this method is especially appropriate for sustainable agriculture. While among the soil organisms favored by organic fertilization, earthworms have been identified as a key functional group (*Jouquet et al., 2006*). Earthworms have a great ability to consume organic wastes, reducing the volume by approximately 50% and expelling the digested materials as castings, which are useful for soil amendments and may be easily stored for agricultural use (*Tomati, Grapelli & Galli, 1985*). *In situ* earthworm breeding in orchards usually has three important advantages. First, this method can be used to manage a large amount of organic wastes from agriculture. At present, earthworms have attracted a great deal of attention as an efficient and low-cost means of composting organic wastes such as animal wastes and crop residues (*Ndegwa & Thompson, 2001*; *Singh et al., 2008*). They not only reduce organic waste pollution but also improve the environment of rural areas. Second, this method produces a large amount of high-quality compost, known as "vermicompost," which comes from the biological degradation of organic wastes by earthworms (*Chaoui, Zibilske & Ohno, 2003*). Third, earthworm activities improve the soil structure, microbial activity and biodiversity, and soil OM dynamics (*Jongmans, Pulleman & Marinissen, 2001*; *Pulleman et al., 2005*; *Jouquet et al., 2007*; *Bottinelli et al., 2010*; *Bernard et al., 2011*). Furthermore, earthworm activity is also an important factor that controls vegetation dynamics and has a positive influence on plant growth (*Doan et al., 2013*). However, there is still a lack of knowledge about the effects of *in situ* earthworm breeding in orchards on the growth, quality, and yield of fruits.

Thus, the aim of our study was to evaluate the effects of chemical fertilizer, compost, and *in situ* earthworm breeding in orchards on the growth, fruit quality, and yield of papayas and to explore a potential application of organic fertilizer that can not only be used as a substitute for chemical fertilization but also improve papaya yield and quality.

## MATERIALS AND METHODS

### Site description

This study was conducted at Yinghuwan reclamation land, Xinhui district, Jiangmen city (23°N, 113°E), which is located in the southwestern Pearl River Delta in Guangdong Province, China. The area is characterized by a typical subtropical monsoon climate. The average annual precipitation is 1,763 mm, of which approximately 80% falls during the wet season between May and September. The annual effective accumulated temperature is 7,693 °C. The average mean temperature is 23.8 °C, with the lowest and highest monthly mean temperatures in January and July, respectively. The annual solar radiation is 110 kcal
**Table 1 The providing nutrient of compost and *in situ* earthworm breeding.**

| Treatment | pH | Total N (g/kg) | Total P (g/kg) | Total K (g/kg) | Organic matter (g/kg) |
|-----------|------|------|------|------|--------|
| O | 6.12 | 9.58 | 4.23 | 4.03 | 193.22 |
| E | 5.98 | 11.64 | 6.64 | 7.60 | 179.70 |

Notes.
    O, compost; E, *in situ* earthworm breeding.

$cm^{-2}$. The background values for the soil pH, soil organic matter, total nitrogen (N), total phosphorus (P), total potassium, available N, available P, and available potassium are 6.72, 24.26 g $kg^{-1}$, 1.21 g $kg^{-1}$, 0.72 g $kg^{-1}$, 22.29 g $kg^{-1}$, 80.92 mg $kg^{-1}$, 62.80 mg $kg^{-1}$, and 286.42 mg $kg^{-1}$, respectively.

## Experimental design and treatments

This experiment was conducted in a Hawaiian papaya orchard from March to December of 2008. Hawaiian papaya plants were transplanted on March 31, with a planting space of 3.1 m × 2.7 m. Four treatments were used in our study. These treatments consisted of a control (C), chemical fertilizer (F), compost (O), and *in situ* earthworm breeding (E). All treatments were repeated three times during the experiment. A total of 5 g $plant^{-1}$ urea ($CH_4N_2O$, %$N = 46.3$%) and 100 g $plant^{-1}$ microelement fertilizer was applied to each treatment. Since then, no chemical or fertilizer were applied to C. However, 45 g $plant^{-1}$ urea ($CH_4N_2O$, %N $= 46.3$%), 100 g $plant^{-1}$ phosphate (%$P_2O_5$, $P = 12$%), 500 g $plant^{-1}$ compound fertilizer were applied in the F treatment. The O was prepared by using cow manure. A total of 10 kg $plant^{-1}$ cow manure was applied to the O treatment. The E field pattern can be found in Fig. 1. An earthworm bed (length: 16 m, above width: 40 cm, below width: 60 cm, and height: 30 cm) was prepared approximately 50 cm from the papaya plant in each plot. We added 4.858 kg $m^{-3}$ organic wastes which produced in the process of producing beer to the bottom of the bed, and then we put earthworms (*Eisenia fetida*) into it at a density of 8 g per $m^2$. Next, we put rice straw and sun shading net on the bed, and water and organic wastes were added regularly so that we could provide a better environment for the earthworms' growth and reproduction. The providing nutrients of O and E were listed in Table 1.

On April 28, 5 g $plant^{-1}$ urea ($CH_4N_2O$, %$N = 46.3$%) was added to each plot. On May 24, 100 g $plant^{-1}$ compound fertilizer (%N-%P-%K $= 15$%-15%-15%) was applied to the F treatment, and 1 kg $plant^{-1}$ cow manure was applied to the O treatment. The chemical fertilizer and compost was applied at the base of the papaya plants. On June 12, the application of F and O was the same as that on May 24. On July 4, 40 g $plant^{-1}$ urea and 100 g $plant^{-1}$ phosphate (%$P_2O_5$, $P = 12$%) were applied to the F treatment, and 2 kg $plant^{-1}$ cow manure was applied to the O treatment. On August 25, 100 g $plant^{-1}$ compound fertilizer was applied to the F, and 3 kg $plant^{-1}$ cow manure was applied to the O. Simultaneously, 100 g $plant^{-1}$ microelement fertilizer was applied to each treatment. On September 12, 200 g $plant^{-1}$ compound fertilizer was distributed over the F, and 3 kg $plant^{-1}$ cow manure was distributed over the O. The total amounts of N, P, K, organic matter and microelements that were included in the fertilizer application for each treatment were analyzed (Table 2).

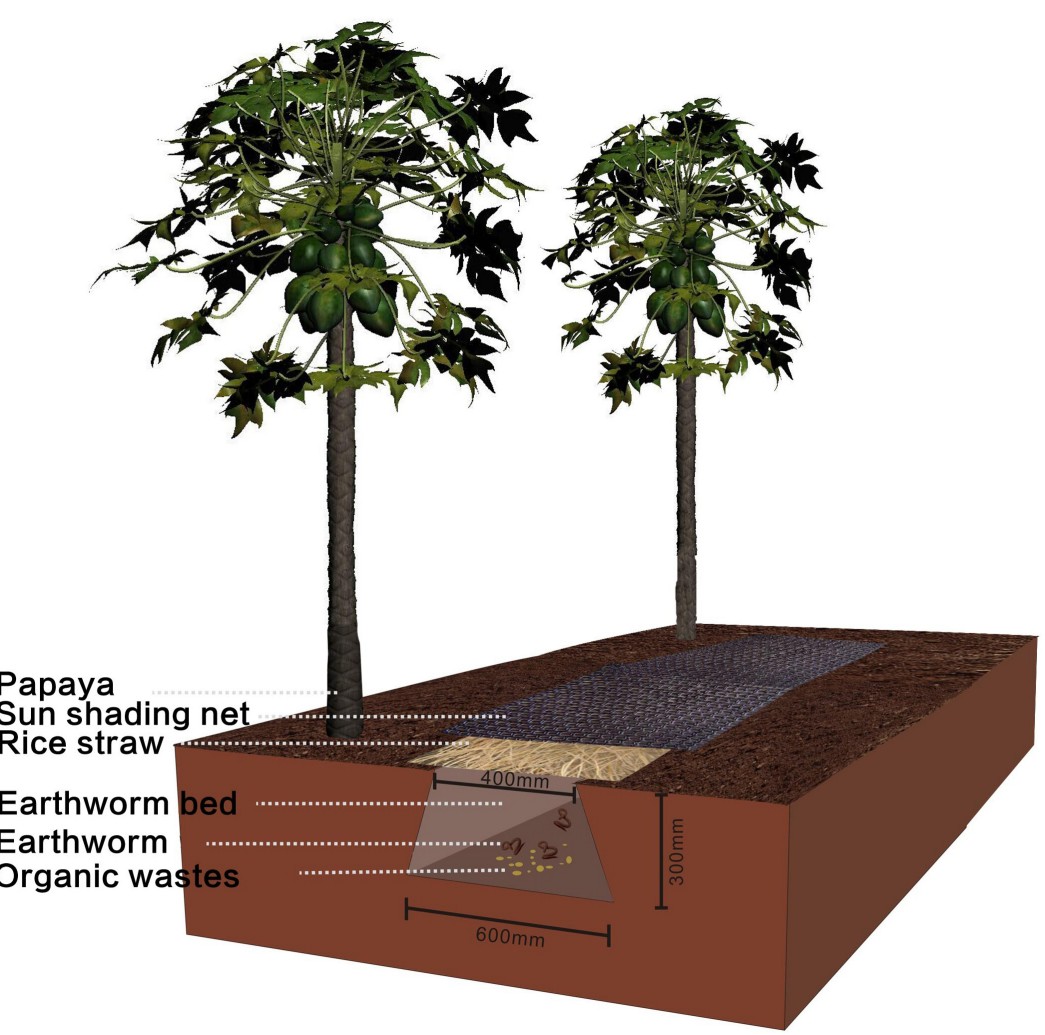

Papaya
Sun shading net
Rice straw
400mm
Earthworm bed
Earthworm
Organic wastes
300mm
600mm

**Figure 1  The E field pattern in a papaya orchard.** E: *in situ* earthworm breeding.

**Table 2  The total amounts of N, P, K, organic matter and microelements in the fertilizer application by human for each treatment.**

| Treatment | Total N (g) | Total P (g) | Total K (g) | Organic matter (kg) | Microelements (g) |
|-----------|-------------|-------------|-------------|---------------------|-------------------|
| C | 2.32 | 0.00 | 0.00 | 0.00 | 50.00 |
| F | 110.88 | 102.00 | 90.00 | 0.00 | 50.00 |
| O | 98.12 | 42.30 | 40.30 | 1.93 | 50.00 |
| E | 2.32 | 0.00 | 0.00 | 0.00 | 50.00 |

**Notes.**

C, control; F, chemical fertilizer; O, compost; E, *in situ* earthworm breeding.

## Measurements

Measurements of the plant height (cm) and stem diameter (cm) were recorded for seven plants from each replication at 38, 55, 76, 94, and 127 days after transplanting the papayas plants. The ripen fruits of five plants from each replication that growing trend consistent were chosen to observation the fruit quality. Quality parameters such as the total acid and soluble-solid content were determined in accordance with the *AOAC (1989)*. The vitamin C content was assessed as described by *Bessey & King (1933)*. Fifty grams of papaya flesh was well homogenized with 50 mL of 2% (w/v) oxalic acid by using a kitchen blender, 20 mL of homogenate was diluted to 50 mL with 2% oxalic acid, and 10 mL of the solution was titrated with 2,6-dichlorophenolindophenol solution until it appeared pink in color.

The total sugar content was measured as follows: 1 g of fleshy tissue was ground in 5 mL of ethanol, and the mixture was then centrifuged at $12,000 \times$ g for 10 min at 4 °C. After that, 0.1 ml of ethanol extract was mixed with 1 ml of 2 g/L anthrone in 706 g/L $H_2SO_4$. The mixture was incubated at 100 °C for 15 min and cooled in a water bath, and the total sugar content was determined at 625 nm. The protein content was measured according to a method described by *Bradford (1976)* with bovine serum protein as the standard, and the results were expressed in mg $g^{-1}$. Electrical conductivity was closely related with fruit storing time, and it was measured by using an Orion Star Plus pH meter (Thermo Fisher Scientific Inc., Singapore). Firmness of 25 fruit samples from each replicate was determined with a texture analyzer (KM-1; Stable Micro Systems, Surrey, UK) with a 2 mm diameter stainless steel probe. The fruits were tested equatorially at their maximum diameter with a cross-head speed of 50 cm $min^{-1}$. The force was expressed in Newtons (N). The fruit numbers were counted at harvest. The individual fruit weights and total yields were measured from the fresh weights of the fruits.

## Statistical analysis

Experimental data were evaluated by analysis of variance (ANOVA), and significant differences between the means of three replicates ($p \leq 0.05$) were determined by Duncan's multiple range tests with SPSS 13.0 for Windows. All figures were created in Origin version 8.

## RESULTS

### Plant growth

Different fertilizer applications significantly increased the plant height and stem diameter growth parameters, and these positive effects were strengthened over time (Fig. 2). For the plant height, a significant difference was found only between the E and the C treatments on June 10 (55 days after planting), and on July 3 (76 days after planting), the plant heights of the F and E treatments were significant higher than that of the C treatment from July 3 ($p \leq 0.05$) (Fig. 2B). On August, the plant heights of the E, F, and O treatments were 195.70, 188.70, and 172.60 cm, respectively, which were 24.70%, 20.23%, and 9.98% higher than the C treatments, respectively (Fig. 2B). The stem diameters showed similar increasing trends under different fertilizer applications (Fig. 2A). On Aug 24 (127 days after planting),

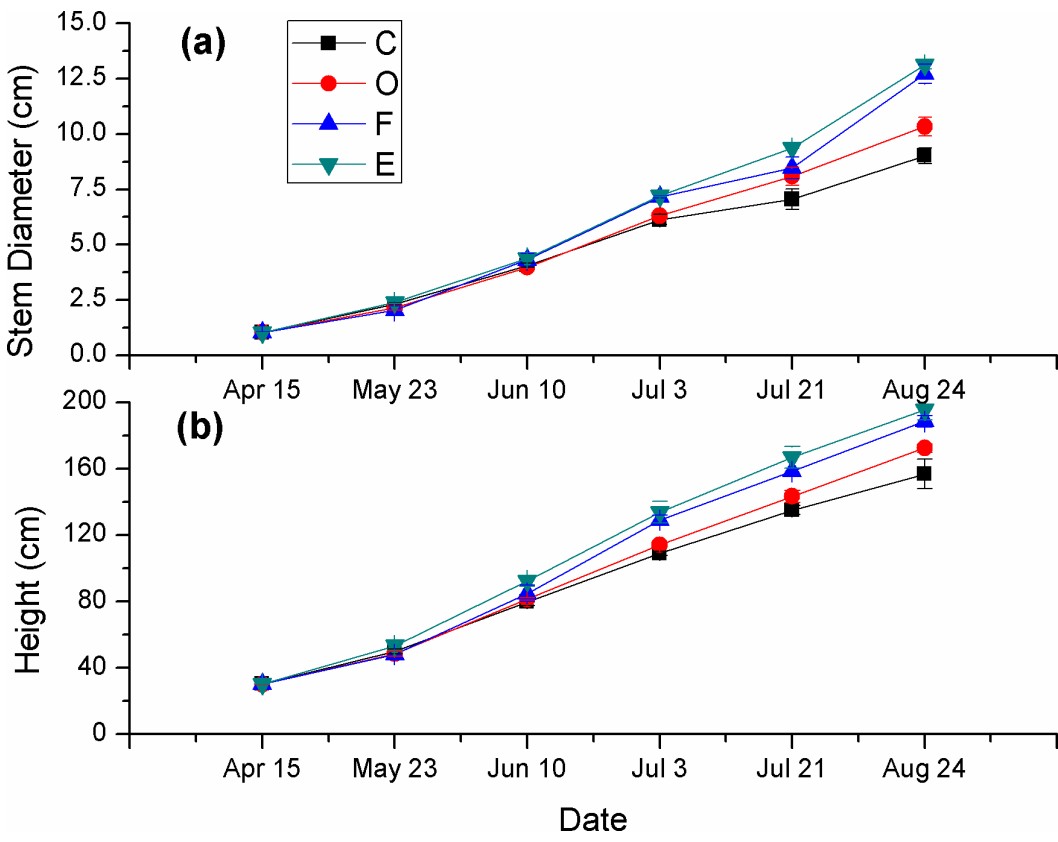

**Figure 2** Plant heights and stem diameters under different fertilizer applications (mean ± standard error, $n = 3$). C, control; O, compost; F, chemical fertilizer; and E, *in situ* earthworm breeding.

the five treatments were also ordered E > F > O > C, and there was a significant difference between all treatments ($p \leq 0.05$).

## Fruit quality

There were significant differences of different fertilizer applications on fruit quality parameters such as the soluble-solid, sugar, vitamin C, and protein content ($p \leq 0.05$). The soluble-solid content in each of the four treatments was ordered E > F > O > C, and this parameter was markedly improved in the E treatment ($p \leq 0.05$) (Fig. 3A). The soluble-solid content in the E treatment was 12.96%, 18.22%, and 28.22% higher than the content of the F, O, and C treatments, respectively ($p \leq 0.05$). The sugar content in the E (8.18%) treatment was also clearly increased, at 5.68%, 31.09% and 19.21% higher than the sugar in the F, O, and C treatments, respectively (Fig. 3B). The vitamin C content in the E treatment was 132.95 mg kg$^{-1}$, which was slightly higher than that of the C treatment ($p \leq 0.05$) (Fig. 3C). The protein content in the E (2.45 g kg$^{-1}$) was also increased; it was 11.36% and 21.89% higher than the content of the F and C treatments (significantly different with $p \leq 0.05$) (Fig. 3D).

The total acid, electrical conductivity and firmness were also affected by the different treatments. The total acid was dramatically decreased in the E treatment ($p \leq 0.05$)

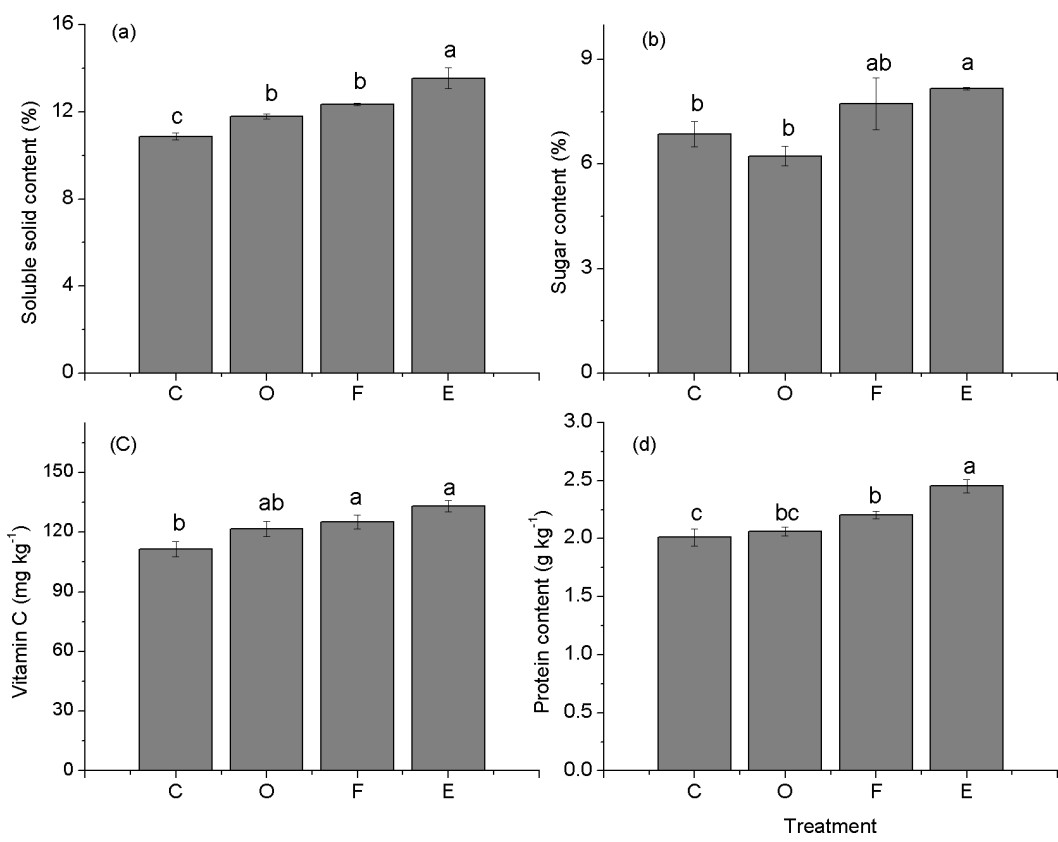

**Figure 3** **Soluble-solid, sugar, vitamin C and protein content of papaya fruit under different fertilizer applications (mean ± standard error, $n = 3$).** Different letters indicate significant differences between treatments at $p \leq 0.05$. C, control; O, compost; F, chemical fertilizer; E, *in situ* earthworm breeding.

(Fig. 4A) and was reduced by 44.28%, 46.86% and 65.31% compared with that of the C, O and F treatments, respectively. The electrical conductivity in the O and E treatments was obviously lower than that of the C ($p \leq 0.05$); they dropped by 31.88% and 28.26% relative to the C treatment (Fig. 4B). The fruit firmness was significantly enhanced in the O, but it decreased in the F treatment ($p \leq 0.05$) (Fig. 4C).

## Fruit yield

The E treatment significantly enhanced the fresh weight per fruit, the fruit number, and the total yield (Table 3). The individual fruit weights for the four treatments were 373.48 (F), 359.17 (E), 299.47 (O), and 241.92 g (C). Compared with the C treatment, the F and E treatments were increased by 54.38% and 48.47%, respectively. In addition, the fruit numbers and total yields were also significantly increased in the F and E treatments, and the E was higher than the F. The quantities of fruits in the F and E treatments were 39.86% and 47.59% higher than that of the C treatment. The total yields of the F and E treatments were 116.60% and 120.62% higher than that of the C treatment, and the E yield was improved by 1.85% relative to the F treatment.

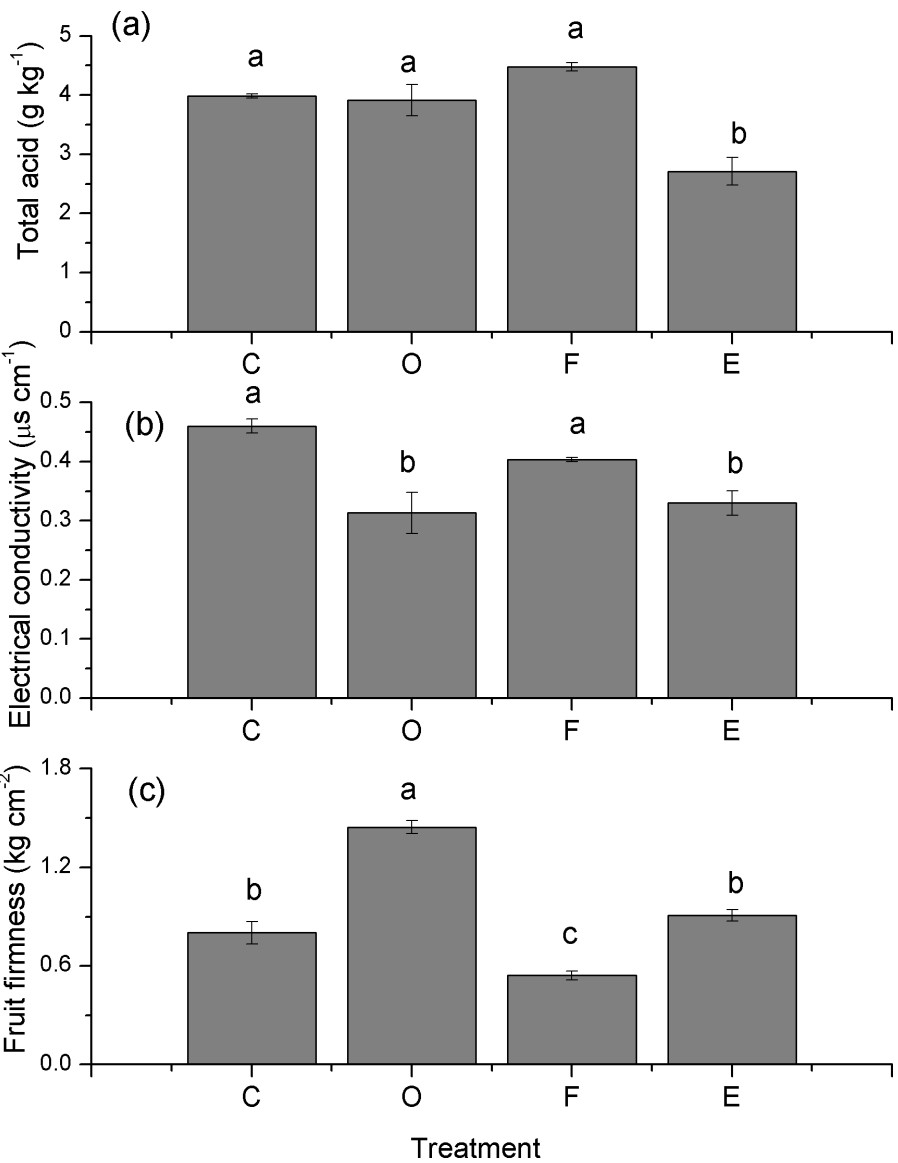

**Figure 4** **Total acid, electrical conductivity and firmness of papaya fruit under different fertilizer applications (mean ± standard error, $n = 3$).** Different letters indicate significant differences between treatments at $p \leq 0.05$. C, control; O, compost; F, chemical fertilizer; E, in situ earthworm breeding.

## DISCUSSION

### Effects of different fertilizer applications on plant growth

In this study, the E treatment significantly promoted papaya plant growth than the treatment of C and O, and this treatment exhibited the highest plant heights and stem diameters of the four treatments over the whole growth period. Our result is consistent with other studies on earthworms in aboveground plant communities (*Piearce, Roggero & Tipping, 1994*; *Wurst, Langel & Scheu, 2005*). The following mechanisms may be related to the results. First, the earthworm activities in the E treatment can improve the soil structure (such as the porosity) and increase the soil nutrients, and thus they provide a better

**Table 3** Effects of different fertilizer applications on the papaya fruit number, weight, and total fruit yield of Hawaiian papaya.

| Treatments | Individual papaya weights (g) | No. of fruits/plant | Total fruit yield (kg hm$^{-2}$) |
|---|---|---|---|
| C | 241.92c | 18.67b | 4835.51c |
| O | 299.47b | 21.44b | 6927.87b |
| F | 373.48a | 26.11a | 10477.19a |
| E | 359.17a | 27.56a | 10671.04a |

Notes.

The means within the same letter are not significantly different according to Duncan's multiple range test at $p \leq 0.05$.

C, control; O, compost; F, chemical fertilizer; E, *in situ* earthworm breeding.

root growth medium. *Derouard et al. (1997)* found that earthworms significantly affect soil aggregation and water infiltration. *Lee (1985)* also noted that earthworms alter the physical, chemical, and biological properties of soil, which can in turn modify the plant growth. In this study, we found that the nutrient content such as total N, P, and K in E treatment was higher than these in O treatment (Table 1). This suggested that E treatment provided a better soil nutrient for the papaya plant. Second, the plant-growth hormones included in the fresh earthworm casts stimulated papaya growth. Numerous studies showed that earthworm casts contain plant-growth-regulating materials such as humic acids (*Senesi, Saiz-Jimenez & Miano, 1992*; *Masciandaro, Ceccanti & Gracia, 1997*; *Atiyeh et al., 2002*) and plant-growth regulators such as auxins, gibberellins, and cytokinins (*Krishnamoorthy & Vajrabhiah, 1986*; *Grappelli, Gallli & Tomati, 1987*; *Tomati et al., 1990*), which contribute to increases in plant growth for many crops (*Atiyeh et al., 2002*). Thus, our study suggested that *in situ* earthworm breeding in orchards can result in better plant growth.

## Effects of different fertilizer applications on the fruit quality

Soluble-solid, sugar, vitamin C, and protein content are very important parameters of fruit nutrition. Increases in the content of these indices can indicate the enhancement of fruit quality. During this field experiment, the E treatment significantly improved the fruit quality because it increased the soluble-solid, sugar, vitamin C, and protein content. The primary reason may be the presence of earthworm casts, which are also known as vermicompost, in the E treatment. Vermicompost could improve the fruit quality, and our results are consistent with previous studies. For example, *Premuzic et al. (1998)* reported that the fruits of tomatoes grown on organic vermicompost substrates contained significantly higher vitamin C than those grown in hydroponic media. *Gutiérrez-Miceli et al. (2007)* suggested that the addition of sheep manure to vermicompost decreased the titratable acidity and increased the soluble and insoluble solids in tomato fruits, compared with those harvested from plants cultivated in unamended soil. The beneficial effects of vermicompost utilization for improving the fruit quality in other horticulture settings have also been reported (*Tomati, Grapelli & Galli, 1987*; *Hidalgo, 1999*; *Saciragic & Dzelilovic, 1986*).

Moreover, the total acid, electrical conductivity, and fruit firmness were another three important indicators of fruit quality. The decreasing total acid content denoted an improvement in fruit flavor, the lower electrical conductivity indicates a longer period of fruit storage, and the higher fruit firmness represents easier storage. In this study, the

total acid and electrical conductivity decreased, but the fruit firmness was increased in the E treatment. Therefore, the E treatment that was incorporated into soil could effectively improve the fruit quality.

### Effects of different fertilizer applications on the fruit yield

Our results suggested that the F and E treatments significantly improved the papaya yield. Several field studies have also found significant increases in fruit yields under earthworm inoculation and vermicompost application (*Goswami, Kalita & Talukdar, 2001*; *Gutiérrez-Miceli et al., 2007*; *Fragoso et al., 1997*). *Goswami, Kalita & Talukdar (2001)* observed that vermicompost addition rates of 0, 20, 30, and 40 t ha$^{-1}$ produced tomato yields of 114, 138, 163, and 192 t ha$^{-1}$ in comparison with the inorganically fertilized tomatoes that received 56 t ha$^{-1}$. *Pashanasi et al. (1996)* found that plant production was significantly increased by 36% following earthworm inoculation into a traditional low-input rotation. The role of earthworms in enhancing plant production depends on the synlocalization and the synchronization of their activities with the period and sphere of active root growth and nutrient demand. Most earthworm species release significant amounts of assimilable nutrients that can be supplied to the plants that grow in their casts (*Syers, Sharpley & Keeney, 1979*; *Lavelle et al., 1992*). Earthworm activities promote the intense mineralization of soil, releasing considerable quantities of mineral N, P, and K. This process may stimulate the plant growth and improve the fruit crop yield correspondingly. As the organic material is processed by the digestive systems of earthworms, vermicomposting differs from conventional composting. The higher N, C, P, K, Ca, and Mg availability in vermicompost implies that it has a function as a slow release source of plant nutrients (*Chaoui, Zibilske & Ohno, 2003*).

However, can the E treatment perform better than the F treatment in improving the total yield? In a previous study, 7.5 t ha$^{-1}$ vermicompost was added to a treatment that increased the marketable fruit yield up to 58.6% relative to that of the inorganic fertilizer treatment (*Singh et al., 2008*). By contrast, the earthworms in the E treatment in our study were added only once, and the density of earthworms was only 8 g m$^{-2}$. Therefore, we extrapolated that if the density of earthworms was increased in a proper range, and it reached an appropriate standard, the effect of the E treatment in terms of improving the fruit yield would be much higher than that of the F treatment. Meanwhile, the price of buying the earthworms are low, it decreased the cost of agriculture. For example, in our study the cost of F and E treatment was 22.19 USD ha$^{-1}$ and 15.29 USD ha$^{-1}$, respectively. The cost of E was lower than this in the F treatment, although the preparation of the earthworm bed in E treatment was complicated. It can be seen as one of the limitations of earthworm breeding. Also, we could not determine if the E treatment would be suitable for any type of papaya fields. This issue needs be to studied further.

## CONCLUSION

This study evaluated the effects of C, O, F, and E treatments on the plant growth, fruit quality, and yield of papayas. Of the four treatments, the E treatment provided the best medium for plant growth. The present study revealed that the E treatment was quite useful

in field-grown papaya for conferring higher fruit quality and total yield. Generally, the incorporation of the E treatment with soil could significantly improve the growth, fruit quality, and yield of papaya in comparison with the C and O treatments. The E treatment could be used as an effective substitute for chemical fertilizer. Furthermore, the E treatment could be a potential organic fertilizer application because it not only improved the quality and yield of fruit but also reduced the amount of organic waste in agriculture as a result of earthworm activity.

### Funding

This research was supported by the Guangdong Industry-University-Research Institute Cooperation Project (2010B090400453 and 2015A090905007), the National Science & Technology Pillar Program of China (2012BAD14B16-04), the Guangdong Province Science and Technology Program of Guangdong Province, China (2012B020310005 and 2015B090903077). The funders had no role in study design, data collection and analysis, decision to publish, or preparation of the manuscript.

### Grant Disclosures

The following grant information was disclosed by the authors:
Guangdong Industry-University-Research Institute Cooperation Project: 2010B090400453, 2015A090905007.
National Science & Technology Pillar Program of China: 2012BAD14B16-04.
Guangdong Province Science and Technology Program: 2012B020310005, 2015B090903077.

### Competing Interests

The authors declare there are no competing interests.

### Author Contributions

- Huimin Xiang, Jia-en Zhang, Lei Guo and Benliang Zhao conceived and designed the experiments, performed the experiments, analyzed the data, contributed reagents/materials/analysis tools, wrote the paper, prepared figures and/or tables, reviewed drafts of the paper.

### Data Availability

The raw data has been supplied as a Data S1.

### Supplemental Information

Supplemental information for this article can be found online at http://dx.doi.org/10.7717/peerj.2752#supplemental-information.

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
