# Peer review of "In situ earthworm breeding in orchards significantly improves the growth, quality and yield of papaya (Carica papaya L.)"

_PeerJ, doi:10.7717/peerj.2752_

## Round 0.1 · original submission · Major Revisions

We now have received three reviews of your submitted paper. Two reviewers have suggested minor revisions, however, third reviewers suggested major revisions. Please revise your paper accordingly and pay particluar attention to the discussion section as suggested by reviewer 1 and 3.

·

Basic reporting

Actual p-values must be provided throughout, either as a table or directly in the text.

Experimental design

Some essential information is missing from the Method section. In particular 1) the spatial arrangement of the different treatments (the distribution should be randomised), 2) the origin of the papaya plants used (this is to ensure that no bias was introduced at the inception of the work) and 3) how much and what type of waste was included in the earthworm breeding treatment.

Validity of the findings

The number of replicates (3 per treatment) is rather low. Therefore, the findings must be taken with caution. This needs to be further acknowledged in the discussion and I think it would be good to include the number of replicates in the legends of the figures.

Additional comments

Line 15. Were these differences significant? if so add, ‘with significant differences between all treatments’.

Line 16. remove ‘the’ before total soluble-solid, …. Presumably these were all measured from the fruit as well?

Line 27. Why is it important? volume exported? volume produced? area of cultivation?

Line 31. Would ‘unreliable’ be a better term than ‘unstable’ here?

Line 66. replace ‘valid’ by ‘potential’.

Lines 77-79. This is difficult to read and a source of errors. It says that pH=6.72 g kg-1

Line 82. How big/old were the plants at the start of the experiment? Under which conditions have they been grown prior to the experiment? Were they randomly distributed to the different treatments with regards to their size?

Line 90. What organic waste was used in the EB treatment and how much is 'sufficient organic waste'?

METHOD general: How were the different treatments arranged in space? Was there any randomisation of the treatments?

Line 118. add ‘plants’ at the end of the sentence.

Line 131. Why is electrical conductivity important to measure? The other measurements are obviously important. This one less so. You give this information in the discussion, but I think it would make sense to mention here why you measured this.

FIGURES general. Please indicate the number of replicates for each treatment in the legends

RESULTS general: please provide F and p-values for ANOVAs, either in a table or directly in the text at the end of each sentence

Line 149. differences would have likely increase if data had been recorded after August 24. So I am not sure that ‘peaked’ is the correct term here.

Line 163. Were all these differences significative?

Line 226. I thought the experiment lasted for 127j. This is not quite the same as one year. Or did you mean one season? Having said that I actually cannot find the information of when exactly were the fruits harvested. Was it just after the final height and stem measurements or was it at a later date?

Line 233. add ‘to’ before vermicompost.

Line 266-269. I am not sure how you can extrapolate this. I agree that a higher density of earthworms is likely to increase the observed positive effect, but I think the sentence needs rephrasing because the beneficial effect of earthworm will probably plateau at some point. It is not possible to affirm that increasing the biomass of earthworm will increase the observed benefits in a linear fashion, which is almost implied in this sentence.

DISCUSSION general. With different concentrations of fertiliser used and different densities of earthworms, the results may have been different. It would be good in the discussion to put into perspective the cost of using FL or EB at the presented concentrations/densities and discuss the practicality of generalising EB as it was used here. Presumably the fact that earthworms are added at the beginning and no top up was required is a clear economic advantage. But buying the earthworms may be expensive? Can EB be prepared in any type of papaya fields? Are there any potential limitations? etc.

·

Basic reporting

No Comments

Experimental design

The submission clearly defined the research question, methods were described, and it accord with the demand of Peer J.

Validity of the findings

No Comments

Additional comments

The manuscript compared the effects of four fertilizer applications—control (CK), chemical fertilizer (FL), compost (CM), and in situ earthworm breeding (EB)—on the growth, quality and yield of papaya (Carica papaya L.). The results indicated the in situ earthworm breeding treatment performed better than conventional compost and chemical fertilizer treatments. The results may provide guidance in agriculture production.
The manuscript suggests publish, but there are still some small mistakes and some data maybe add to support the discussion.

1. Line 85-87, the sentence “The CK was added at 5……applied afterwards” can delete, and add “Since then, no chemical or fertilizer were applied to CK” after the first sentence at line 95.
2. Line 118, which kind of papaya were picked to analyze the quality parameters?
3. Line 147-148, the plant heights of the FL and EB treatments were significant higher than that of the CK treatment. It seems from Jun 10 ? or when?
4. Line 145, Fig. 2a revised as Fig. 2. And Line 148 add “(Fig. 2b)”, line 153 Fig. 2b revised as “Fig. 2a”. This maybe the mistake of arrange of Fig. 2
5. Line185, add “Total acid” after “Fig. 4”
6. About the discussion of 4.1, if the authors conserved the soil, they had better to analyze the initial and the last soil nutrient (N, P, K, organic matter) and soil structure, which could help to explain the mechanism of plant growth.

Reviewer 3 ·

Basic reporting

Remove or change abbreviations (C= control, F= chemical fertiliser, O= organic fertiliser/compost, E= earthworm breeding.

L43-50 This is confusing as it talks about vermicomposting and then you go on to state how it is different to earthworm breeding. In the discussion you then state that some of the benefits are a result of vermicompost. Need to be clear and consistent. Maybe you are trying to say that the advantage of having earthworm breeding on site is that you can use wastes you generate yourself, rather than buying in vermicompost? Need to modify introduction accordingly.

Experimental design

Need to provide more details on each of the methods used

L84 What and how much chemical fertiliser was applied? Move from L96.
L87 how much cow manure? Equivalent nutrient applied? Move from L97
L90 What kind or organic wastes, how much?
L92 What kind or organic wastes, how much?
L101 How was chemical fertiliser and compost applied? At the base of the plants?
L102 If urea and microelement fertiliser was applied to all plots this needs to be clearly stated (also in the abstract).

Figure 1
Is the idea that the Papaya roots will grow into the sites of earthworm breeding?

Table 2
What were the levels of nutrients in the earthworm breeding/vermicompost. The organic wastes will be providing nutrients.

Validity of the findings

Plant height and stem diameter do not look significantly different between earthworm breeding and chemical fertilizer (this is stated in L245), will need to reword manuscript and combine these two sections accordingly. Since there was no significant difference between these two treatments I would assume that effect on plant growth is more related to the nutrients available to the plants so need measure nutrient content in earthworm breeding bins and to rewrite discussion L211-215.

Additional comments

Specific comments:
L32 rephrase or explain ‘it is not environmentally friendly’
L33 how does reducing the C:N ratio impair soil fertility?
L33 remove ‘seriously’ and give examples.
L47 remove ‘all’
L76 what is the average mean temperature?
L226 this was not a one year study it ran from March to December.
L260 why slow release source?

---

## Round 0.2 · accepted · Accept

Please note minor corrections as below,

Line 15 'Then the fertilizer applications of these four treatments are different from each other.' needs to be written again for clarity.
Line 17 'the plant heights were ordered' meaning is not clear, please consider revising.